# A safety cap protects hydrogenase from oxygen attack

Martin Winkler[1,7], Jifu Duan [1,7], Andreas Rutz[1], Christina Felbek [2], Lisa Scholtysek[1], Oliver Lampret[1], Jan Jaenecke[1], Ulf-Peter Apfel[3,4], Gianfranco Gilardi [5], Francesca Valetti [5], Vincent Fourmond [2], Eckhard Hofmann [6], Christophe Léger [2✉] & Thomas Happe [1✉]

[FeFe]-hydrogenases are efficient $H_2$-catalysts, yet upon contact with dioxygen their catalytic cofactor (H-cluster) is irreversibly inactivated. Here, we combine X-ray crystallography, rational protein design, direct electrochemistry, and Fourier-transform infrared spectroscopy to describe a protein morphing mechanism that controls the reversible transition between the catalytic $H_{ox}$-state and the inactive but oxygen-resistant $H_{inact}$-state in [FeFe]-hydrogenase CbA5H of *Clostridium beijerinckii*. The X-ray structure of air-exposed CbA5H reveals that a conserved cysteine residue in the local environment of the active site (H-cluster) directly coordinates the substrate-binding site, providing a safety cap that prevents $O_2$-binding and consequently, cofactor degradation. This protection mechanism depends on three non-conserved amino acids situated approximately 13 Å away from the H-cluster, demonstrating that the 1st coordination sphere chemistry of the H-cluster can be remote-controlled by distant residues.

[1] Photobiotechnology, Department of Plant Biochemistry, Ruhr-Universität Bochum, 44801 Bochum, Germany. [2] CNRS, Aix-Marseille Université, Laboratoire de Bioénergétique et Ingénierie des Protéines, Institut de Microbiologie de la Méditerranée, Marseille, France. [3] Inorganic Chemistry I, Department of Chemistry and Biochemistry, Ruhr-Universität Bochum, 44801 Bochum, Germany. [4] Fraunhofer UMSICHT, 46047 Oberhausen, Germany. [5] Department of Life Sciences and Systems Biology, University of Torino, Torino 10123, Italy. [6] Protein Crystallography, Department of Biophysics, Ruhr-Universität Bochum, 44801 Bochum, Germany. [7] These authors contributed equally: Martin Winkler, Jifu Duan. ✉email: christophe.leger@imm.cnrs.fr; thomas.happe@rub.de

**[Fe** Fe]-hydrogenases catalyze the reversible reduction of protons to dihydrogen at low overpotential and high turnover rates[1–3]. Their catalytic cofactor consists of a [4Fe-4S]-cluster ([4Fe]$_H$) that is covalently bound to a diiron complex ([2Fe]$_H$) by a bridging cysteine[4]. The two Fe-sites of the [2Fe]$_H$ complex, referred to as proximal (Fe$_p$) and distal (Fe$_d$) according to their distance to the [4Fe]$_H$-cluster, are coordinated by three carbon monoxide (CO) and two cyanide (CN$^-$) ligands[1]. They exhibit strong vibrational absorption signals at discrete infrared frequencies, which respond to changes in redox and protonation states of the H-cluster[5–7]. The diatomic ligands are responsible for the low-spin state of the [2Fe]$_H$ sub-cluster and stabilize it in a configuration that creates an open coordination site at Fe$_d$, allowing the binding and heterolytic splitting of dihydrogen[8,9]. The open coordination site at Fe$_d$ is also the major target of inhibitors such as CO and dioxygen[10–14]. The pending bridgehead amine-group of the azadithiolate-ligand which connects Fe$_p$ and Fe$_d$ in the [2Fe]$_H$-cluster shuttles protons between Fe$_d$ and a nearby located cysteine residue (C367 in CbA5H), which is part of the highly conserved proton transfer pathway[4,6,15–18].

Dioxygen irreversibly damages the H-cluster of most [FeFe]-hydrogenases[10–12,14]. After reaching the active center by diffusion via packing defects[11,19–21], O$_2$ binds to the open coordination site at Fe$_d$, forming a transient adduct which is further transformed by successive protonation and reduction steps[11,14,22,23]. To a limited extent it can be fully reduced to water, partial reduction and protonation however, lead to the production of reactive oxygen species (ROS) that cause H-cluster destruction[11,12,22–26]. To the best of our knowledge, among the [FeFe]-hydrogenases that have been characterized so far, only the recently isolated CbA5H of *Clostridium beijerinckii*, a gram-positive, anaerobic bacterium isolated from soil and feces, has been shown to resist long-term exposure to O$_2$, by reversibly switching from the active oxidized ready state H$_{ox}$ to the O$_2$-protected but inactive H-cluster state H$_{inact}$[27]. This could be demonstrated by monitoring the spectroscopic signature of the enzyme that is repeatedly oxidized in air and reduced by H$_2$ (ref. [27] and Supplementary Figure 1). The H$_{inact}$-state was originally identified as the 'as isolated' form of [FeFe]-hydrogenase DdH from *Desulfovibrio desulfuricans* after homologous expression and aerobic purification[28,29]. H$_{inact}$ provides full protection against O$_2$ prior to the first reductive activation to H$_{ox}$ and can be identified spectroscopically by a characteristic set of IR-signals. However, DdH becomes O$_2$-sensitive after gaining catalytic activity[10,19]. For DdH and some other [FeFe]-hydrogenases, sulfide-addition under oxidizing conditions (either aerobic or anaerobic) also produces the H$_{inact}$-state; the anaerobic formation of H$_{inact}$ is detected in cyclic voltammetry by the early onset of oxidative inactivation[30,31]. Rodríguez-Maciá et al. concluded that H$_{inact}$-formation under oxidizing conditions is the consequence of sulfide-binding to Fe$_d$[30], but this protected state is also formed in CbA5H[27,31] and CpIII ([FeFe]-hydrogenase III from *Clostridium pasteurianum*)[32] in the absence of exogenous sulfide. The sulfide-independent, reversible transition between H$_{ox}$ and the inactive, O$_2$-resistant H$_{inact}$-state is therefore an uncharacterized and intriguing feature of certain [FeFe]-hydrogenases, which significantly enhances their utilization potential (Fig. 1; CbA5H).

In this work, we elucidate the mechanism that protects CbA5H from dioxygen employing a multidisciplinary approach. We describe the structural rearrangement of a polypeptide-loop close to the active site that attributes a 'safety cap' function to the conserved cysteine at position 367. It shields the open coordination site of Fe$_d$ from O$_2$ by enabling the thiol-group to reversibly bind to Fe$_d$, thus providing an intrinsic source for the protective sulfide-ligand.

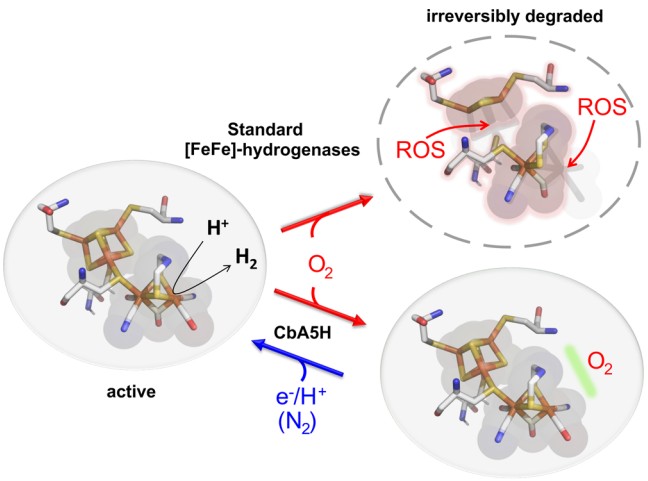

**Fig. 1 The unusual O$_2$-resistance of [FeFe]-hydrogenase CbA5H.** The H-cluster of CpI and other [FeFe]-hydrogenases is irreversibly destroyed when exposed to O$_2$. CbA5H is a rare exception, as it reversibly converted into an inactive but O$_2$-protected state (H$_{inact}$), even in the absence of exogenous sulfide. From the O$_2$-protected state, CbA5H can be reactivated by reduction under anaerobic conditions. ROS: reactive O$_2$-species resulting from O$_2$-activation after Fe$_d$-binding. Green bar: unknown feature or mechanism, protecting the H-cluster of CbA5H from O$_2$-attack in H$_{inact}$.

## Results

**Crystal structure of O$_2$-exposed CbA5H$^{WT}$.** To uncover the structure of the O$_2$-protected enzyme state, we crystallized wild-type CbA5H (CbA5H$^{WT}$) under aerobic conditions (CbA5H$^{air}$) (for details on the crystal and overall structural features see Supplementary Discussion 1, Supplementary Figs. 2–5 and Supplementary Tables 1, 2). The overall structure of the H-cluster domain of CbA5H$^{air}$ is similar to that of standard [FeFe]-hydrogenases like CpI and DdH[4,33]. Interestingly, it embeds the H-cluster with nearly full occupancy (>90%) (Supplementary Table 2); this stability of the active site in the air-exposed crystal contrasts with the observation that the H-cluster of standard [FeFe]-hydrogenases is destroyed under air[26]. Since we could not crystallize CbA5H under anaerobic conditions, we compared the structures of CbA5H$^{air}$ and CpI, a "standard" [FeFe]-hydrogenase. The structure of CbA5H$^{air}$ shows localized structural differences with anaerobically purified CpI (PDB ID: 4XDC)[9] (Supplementary Figure 3). The peptide-loop that spans T365, S366, and C367 in CbA5H (hereafter "TSC-loop", corresponding to T297, S298, and C299 in CpI) is shifted from the conformation observed in the structures of CpI, DdH, or HydA1 (Fig. 2e and Supplementary Figure 5). The bulky side chain of the strictly conserved tryptophan 371 (W303 in CpI), adjacent to the TSC-loop, exhibits a different conformation in CbA5H$^{air}$, hinging away from the loop. The alpha helix carrying this residue is slightly shifted away from the H-cluster. Consequently, the orientation of the residue of the conserved cysteine C367 in CbA5H$^{air}$ is different from that of the corresponding side chain in standard [FeFe]-hydrogenases (Fig. 2d, e and Supplementary Figs. 3 and 5). This observation is consistent with a recently published hypothesis according to which the loop that bears C367 in CbA5H may be flexible enough to allow the binding of the cysteine to Fe$_d$[31]. The strictly conserved cysteine residue is involved in long-range proton transfer to and from the H-cluster (Supplementary Figure 6)[4,16,18]. In CbA5H$^{air}$, the distance between the sulfur atom of C367 and Fe$_d$ is only 3.1 Å, compared to 5.9 Å in CpI (Fig. 2e)[9] indicating bond-formation. The length of this C367-Fe$_d$ bond is longer than the average length of 2.4 Å

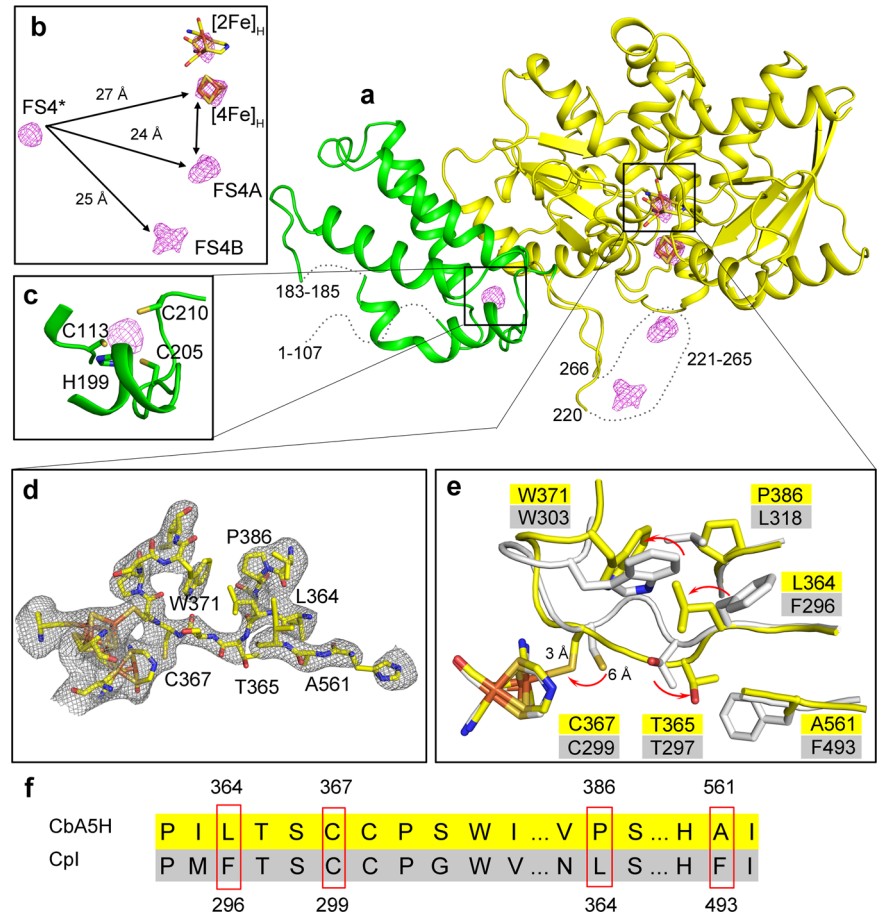

**Fig. 2 X-ray structure of CbA5H$^{WT}$ crystallized under aerobic conditions (CbA5H$^{air}$). a** Cartoon structure of CbA5H$^{air}$ (PDB ID: 6TTL, chain A; see Supplementary Figure 2 for the homodimer). H- and F-domain are colored in yellow, the N-terminal SLBB domain (soluble ligand binding β-grasp)[51] is presented in green. **b** Anomalous electron density map, depicting the positions and distances between iron-sulfur clusters, including the H-cluster (stick model), the two F-clusters (FS4A and FS4B) and the additional N-terminal cluster within the SLBB domain (FS4*) located more than 20 Å away from any other cluster. **c** Potential cluster coordination site within the SLBB domain, consisting of 3 cysteine and one histidine ligand. **d, e** CbA5H$^{air}$ exhibits a characteristic structural displacement of the peptide loop containing C367 (TSC-loop) nearby the [2Fe]$_H$-cluster. **d** The omit map (Fo-Fc) was contoured at 2 σ (see Supplementary Fig. 17 for a more detailed structural comparison between the loop regions of CpI and CbA5H$^{air}$). **e** Structural alignment, depicting conformational differences between CbA5H$^{air}$ (yellow) and CpI (white). Panels **d** and **e** focus on the [2Fe]$_H$-cluster and side chains of amino acids which influence anaerobic inactivation, O$_2$-resistance, and H$_{inact}$ formation. **f** Selected parts of an amino acid sequence alignment of CbA5H$^{WT}$ and CpI, including the polypeptide positions that influence TSC-loop reconfiguration and H$_{inact}$ state formation in CbA5H$^{WT}$.

for a covalent bond between iron and sulfur[34], but short enough to prohibit the insertion of any molecule (including H$_2$ and O$_2$) between the sulfur atom and Fe$_d$. This observation suggests that the structure of CbA5H$^{air}$ is that of an inactive state and explains why the H-cluster is stable under air. That the infrared spectrum of air-exposed CbA5H$^{WT}$ (Fig. 3c and Supplementary Figure 1) shows the typical signature of the H$_{inact}$ state suggests that the X-ray structure of CbA5H$^{air}$, with C367 attached to Fe$_d$, is actually that of H$_{inact}$.

**The crucial role of C367 for H$_{inact}$-state formation.** We compared the properties of CbA5H$^{WT}$ and two site-directed variants in which C367 is replaced with either aspartate or alanine. Below, we demonstrate that each of the three distinctive properties of CbA5H is dependent on the presence of a cysteine at position 367: (1) the non-standard anaerobic oxidative inactivation which occurs at low oxidative potential, (2) the resistance to O$_2$ that results from this inactivation, and (3) the reversible formation of the H$_{inact}$-state upon exposure to O$_2$. In standard [FeFe]-hydrogenases, the replacement of the conserved cysteine in the proton

transfer pathway with aspartate is the only substitution that preserves a significant level of enzyme activity[16,18]. Consistent with previous results with CpI and HydA1, we observed that replacing C367 with alanine renders CbA5H inactive while variant C367D retains 20% of the H$_2$-production activity measured for CbA5H$^{WT}$ and exhibits a shifted pH-optimum (Supplementary Figure 7)[18].

H$_{inact}$ can be accumulated anaerobically by incubating CbA5H$^{WT}$ with oxidants (DCIP or thionine)[27]. Such oxidative treatment can be mimicked in protein film electrochemistry (PFE) experiments by applying high enough electrode potentials[35,36]. The voltammetric response of CbA5H$^{WT}$ (black curve in Fig. 3a, d) resembles that of CpIII[32] and markedly differs from that of all other characterized [FeFe]-hydrogenases[37]. As expected, a negative current is seen at electrode potentials lower than the Nernst potential of the H$^+$/H$_2$ redox couple[38], which reveals proton reduction under reductive conditions. Upon increasing the electrode potential, the current becomes positive, but in contrast to other [FeFe]-hydrogenases, the current strongly decreases above −0.3 V vs SHE (standard hydrogen electrode, all potentials refer to SHE) as a result of anaerobic, oxidative

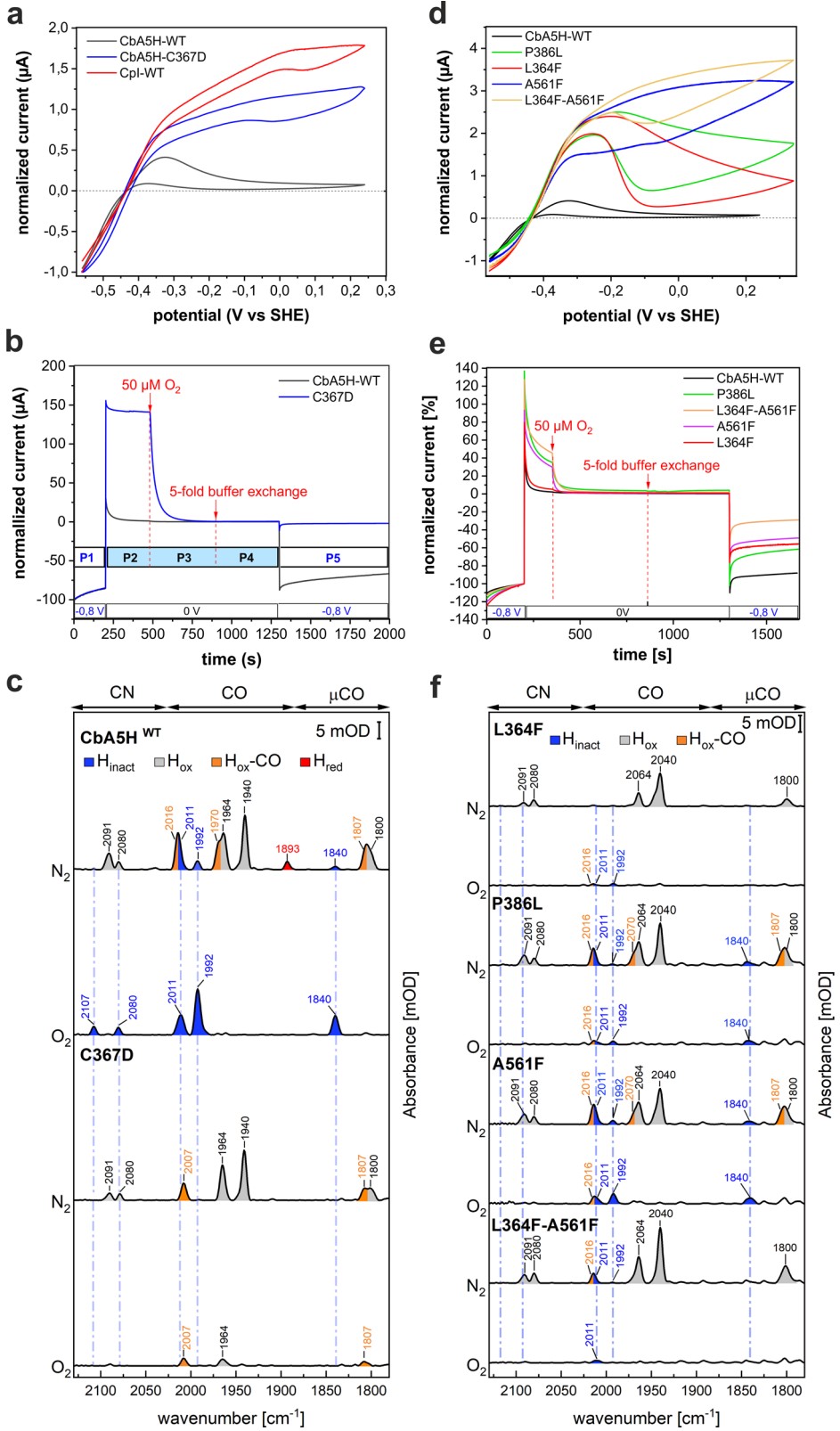

inactivation, as recently observed in another electrochemical investigation of CbA5H[31]. This inactivation at relatively low potential is reminiscent of the shift in the onset of anaerobic inactivation to lower potentials observed for DdH and HydA1 in the presence of exogenous sulfide[30]. When subsequently sweeping the potential down, a faint increase in current, starting below −0.25 V, reveals the beginning of enzyme reactivation, which is complete only when the potential reaches −0.6 V. The C367D exchange causes a striking effect on the voltammetry (blue trace in Fig. 3a): the onset of inactivation occurs at a significantly

**Fig. 3 Electrochemical and spectroscopic features of CbA5H$^{WT}$ and mutagenesis variants. a** Comparison of cyclic voltammograms of CbA5H$^{WT}$ (Cb-WT), CpI (CpI-WT), and Cb-variant C367D (T = 5 °C, pH 7, 1 atm. of H$_2$, scan rate 3 mV/s, electrode rotation rate 3000 rpm, currents normalized at E = −0.56 V). **b** Potential step chronoamperometry of CbA5H$^{WT}$ and C367D (5 °C, pH 7, 1 atm. of H$_2$, 1000 rpm). **P1**: H$_2$-production current at −0.8 V prior to O$_2$-exposure; **P2**: potential step to 0 V; **P3**: injection of 50 μM O$_2$; **P4**: five-fold buffer exchange to re-establish anaerobic conditions; **P5**: potential step to −0.8 V to measure the residual H$_2$-production current. **c** ATR-FTIR-spectroscopy of CbA5H$^{WT}$ and variant C367D prior and after O$_2$-exposure (pH 8). **d** Cyclic voltammograms of CbA5H$^{WT}$ (black) and mutagenesis variants L364F, P386L, A561F, and L364F-A561F (5 °C, pH 7, 1 atm. of H$_2$, 20 mV/s, 3000 rpm, currents normalized at E = −0.32 V except WT, at −0.56 V). **e** Chronoamperometry as in panel B, with CbA5H$^{WT}$ and the same variants as in **d**. **f** ATR-FTIR-spectroscopy (pH 8) of the loop variants, prior, and after O$_2$-exposure. Electrochemical and IR spectroscopic experiments have been repeated for each protein at 3-4 times with consistent results. Source data are provided in a source data file.

higher potential than for CbA5H$^{WT}$, thus restoring a "standard" voltammetric signature. The latter is illustrated for CpI and HydA1 in Fig. 3a and Supplementary Figure 8, respectively: inactivation occurs at a much higher potential (above 0 V) than for CbA5H, and reactivation begins below −0.1 V. The standard oxidative inactivation of [FeFe]-hydrogenases is fully dependent on the presence of chloride ions acting as uncompetitive inhibitors[39]. Inhibition by chloride is also observed for variant C367D, whereas chloride has only minor effects on CbA5H$^{WT}$ (Supplementary Figure 9). Anaerobic inactivation of standard hydrogenases and CbA5H-variant C367D on the one hand, and CbA5H$^{WT}$ on the other hand, therefore, result from distinct mechanisms.

To quantify and compare the O$_2$-sensitivity of CbA5H$^{WT}$ and variant C367D, we employed a PFE chronoamperometric procedure where the enzyme is exposed to O$_2$ under oxidizing conditions, and the level of activity is measured under reductive conditions before and after exposure to O$_2$ (Fig. 3b). The measured catalytic H$_2$-evolution currents were normalized by the initial value recorded at the end of the 1st step at low potential (−0.3 V). Upon switching the potential to 0 V, CbA5H$^{WT}$ instantly inactivates whereas C367D retains >95% of its H$_2$-oxidation activity. Injecting 50 μM O$_2$ into the electrochemical cell fully inhibits the C367D variant. After removal of O$_2$ from the system by a five-fold buffer exchange and subsequently shifting the potential back to −0.8 V, CbA5H$^{WT}$ returns to its original H$^+$ reduction activity, unlike variant C367D, which appears to have been completely and irreversibly inactivated by O$_2$-exposure. These experiments demonstrate that anaerobic inactivation at relatively low potential and resistance to O$_2$-induced damage both depend on C367.

As observed by IR-spectroscopy, exposure of CbA5H$^{WT}$ to air induces a quantitative transition from H$_{ox}$ to H$_{inact}$ (Fig. 3c and Supplementary Figure 1)[27]. In contrast to wild-type enzyme, the O$_2$-treated variants C367D and C367A exhibit none of the IR-vibrational signals characteristic of the H$_{inact}$ state (Fig. 3c and Supplementary Figure 10). Under N$_2$ atmosphere, the C367D variant exhibits strong H-cluster signals which demonstrate a mixture of H$_{ox}$ and H$_{ox}$-CO. O$_2$-treatment of C367D leads, beside a small fraction of the inert H$_{ox}$-CO state (indicative of H-cluster degradation as in ref. [25]), to a fast and nearly complete loss of the H-cluster specific IR-vibrational signals; this suggests that substantial cofactor degradation occurs, as observed for standard [FeFe]-hydrogenases[24–26] (Supplementary Figure 10 and Supplementary Discussion 2 for the corresponding IR-spectroscopy data of variant C367A).

We have therefore shown that (1) anaerobic inactivation, (2) O$_2$-resistance, and (3) aerobic formation of H$_{inact}$ all occur due to the presence of C367 which according to our structure of CbA5H$^{air}$ binds to Fe$_d$ in the H$_{inact}$ state. This saturates the coordination sphere of Fe$_d$ and thus prevents the binding of substrate (H$_2$) and inhibitor (O$_2$) (Fig. 2d–e), rendering the enzyme inactive but protected from O$_2$-induced degradation. That CbA5H is quantitatively locked in the H$_{inact}$ configuration is

not only obvious from the homogenous IR-spectra of O$_2$ treated enzyme but also from the low b-factor values observed for the H-cluster environment in CbA5H$^{air}$ (6TTL) including the fully shifted TSC-loop (Supplementary Fig. 16).

That both, aerobic and anaerobic oxidative conditions induce the formation of H$_{inact}$ suggests that the sensing of oxidative conditions occurs via the two accessory [4Fe-4S]-clusters (FS4B and FS4A in Fig. 2b). They mediate electron transfer between external redox partners and the H-cluster and can be oxidized either by the electrode at high potential or by transferring electrons to molecular oxygen. This reaction with O$_2$ appears to be reversible.

**Kinetics and mechanism of reversible anaerobic inactivation.** We examined the kinetics and potential-dependence of anaerobic (in)activation, which can be quantitatively probed using PFE by potential-step experiments[40]. Figure 4a shows a typical sequence of steps and the resulting change in current. Each initial current peak (e.g., at 50 or 100 s) results from the turnover frequency of the fully active enzyme instantly changing after the potential is stepped up or down; the subsequent slow change in current results from the accordingly slow change in the concentration of the H$_{inact}$ state. The data can be interpreted by assuming various kinetic models, which we tested by fitting them to the corresponding current traces[41]. A model that considers just two species (corresponding to the cysteine being either unbound or bound to Fe$_d$) and resulting in mono-exponential changes in current after each potential step is not satisfactory (green trace in Fig. 4a): the kinetic traces are multiphasic, and any good kinetic model must therefore include the conversions between at least three species (red trace in Fig. 4a). After having analyzed the data recorded at different potentials and pH values, we concluded that the simplest good model is the following, where A means "active", and A$_1$ and A$_2$ are two distinct active species.

$$A_1 \underset{k_{-1}}{\overset{k_1}{\rightleftarrows}} A_2 \underset{k_{react}}{\overset{k_{inact}}{\rightleftarrows}} H_{inact}$$

Fig. 4a shows the fit (red dotted line) of the above model to the chronoamperometric data, from which the values of the four rate constants can be determined at the two potentials used in the experiment. (see Supplementary Notes on the kinetic modeling of anaerobic inactivation of CbA5H and Supplementary Figure 11).

Since the C367D mutation prevents the formation of H$_{inact}$ and gives back CbA5H standard catalytic properties, we assume that the structure of the active form A$_1$ of CbA5H is similar to that of standard hydrogenases, and that the difference between the structures of CpI and CbA5H shown in Fig. 2e illustrates the conformational change that occurs upon the formation of H$_{inact}$ from the active form A$_1$ of the enzyme. The conversion between the active forms A$_1$ and A$_2$ occurs on the time scale of seconds (1/(k$_1$ + k$_{-1}$) =10 s) which implies that there must be a large activation energy barrier to overcome, probably due to a conformational change. In state A$_2$, C367 cannot be bound to Fe$_d$, since A$_2$ is still active (about half as active as A$_1$,

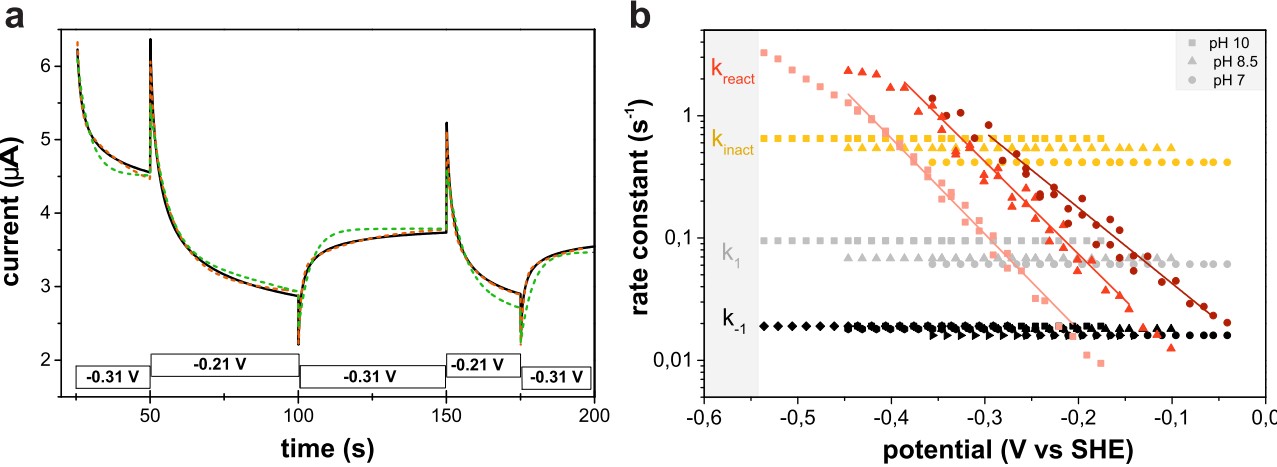

**Fig. 4 Anaerobic conversion between active and inactive CbA5H$^{WT}$ monitored by PFE. a** Chronoamperograms recorded to analyze the kinetics of anaerobic (in)activation of CbA5H$^{WT}$ by stepping between $-0.31$ and $-0.21$ V. The boxes along the abscissa depict the sequence of applied potential steps and the corresponding current response (black line). The plot also shows the best fits of models that consider the interconversion between two (green dotted line) or three (red dotted line) species. Experimental parameters: 5 °C, pH 7, 3000 rpm. **b** Dependence of the rate constants of the "AAI" model (A1 ↔ A2 ↔ H$_{inact}$) on potential and pH, based on the analyses of chronoamperograms recorded for CbA5H$^{WT}$ at pH 10 (squares), 8.5 (triangle), and 7 (circle). Only $k_{react}$ significantly depends on pH and electrode potential. Source data are provided in a source data file.

Supplementary Figure 12). The binding of the cysteine sulfur to Fe$_d$ must therefore occur in the final step (A$_2$ to I). That this bond is labile ($1/(k_{inact} + k_{react}) \simeq 1s$ at high potential) is consistent with the relatively long distance of 3.1 Å between Fe$_d$ and the S-atom of C367 observed in the structure of air-oxidized CbA5H$^{WT}$. The reactivation from H$_{inact}$ to A$_2$ is the only step whose rate constant ($k_{react}$) depends on pH and the applied potential, showing that reactivation is triggered by a reduction step coupled to a protonation. At high potential, the reactivation rate constant $k_{react}$ is lower than the inactivation rate constant $k_{inact}$, locking down the enzyme in the H$_{inact}$ state. The fact that $k_{inact}$ is potential-independent points to a classical "CE" mechanism for step 2, in which one or a series of chemical step(s) ("C") precede(s) proton-coupled oxidation ("E"). Strong coupling implies that the sites of deprotonation and oxidation are very close to one another. We therefore consider likely that this deprotonation occurs from the nitrogen atom of the azaditiolate bridge, or from the bound cysteine, to produce a thiolate ligand.

**Three distal residues determine reversible H$_{inact}$-formation.** The formation of H$_{inact}$ as a result of cysteine binding to Fe$_d$ is a unique property of CbA5H, and yet C367 and the other residues of the shifted TSC-loop are strictly conserved (Supplementary Figure 5). Structural differences in their environment must therefore account for the loop-flexibility that enables the trans-location of C367 in CbA5H$^{WT}$. The most prominent differences between standard hydrogenases and CbA5H$^{WT}$ near the TSC-loop concern the residues at positions 364 (leucine), 561 (alanine) and 386 (proline) (Fig. 2e, f and Supplementary Figure 5): their backbones align with those of their counterparts in standard hydrogenases, but the latter exhibit bulkier residues.

To assess whether the residues at these positions determine the possibility of a conformational change and are responsible for the unique properties of CbA5H, we produced the CbA5H-variants L364F, A561F, P386L and the double-exchange variant L364F-A561F, and characterized them using the same combination of experiments as described above. Only variant P386L shows a reduced H$_2$-evolution activity (55% compared to CbA5H$^{WT}$, Supplementary Figure 7).

The cyclic voltammograms recorded with the four variants show reversible anaerobic inactivation, but at a significantly

**Table 1 Values of $k_1$, $k_{-1}$, $k_{inact}$, $k_{react}$ (the latter at $-0.196$ V, all at 5 °C) for CbA5H$^{WT}$ and variants.**

| | $k_1$ (s$^{-1}$) | $k_{-1}$ (s$^{-1}$) | $k_{inact}$ (s$^{-1}$) | $k_{react}$ (s$^{-1}$) |
|---|---|---|---|---|
| WT (pH 10) | 0.095 | 0.019 | 0.65 | 0.015 |
| WT (pH 8.5) | 0.068 | 0.018 | 0.54 | 0.069 |
| **WT (pH 7)** | **0.061** | **0.016** | **0.42** | **0.17** |
| A561F (pH7) | 0.025 | 0.058 | 0.28 | 0.19 |
| L364F (pH7) | 0.055 | 0.017 | 0.14 | 1.45 |
| P386L (pH7) | 0.049 | 0.051 | 0.15 | 1.50 |
| A561F-L364F (pH 7) | 0.017 | 0.067 | 0.13 | 2.09 |
| Accuracy | 8% | 28% | 7% | 6% |

Rate constants were obtained by analyzing kinetic data of anaerobic inactivation such as those shown in Fig. 4a with the "AAI" model (Supplementary Fig. 14 shows the dependence of $k_{react}$ on E). The CbA5H$^{WT}$ reference for the kinetic parameters of the variant proteins, measured at pH7 is marked in bold. The % accuracy is our estimate of the maximal error for the determination of each rate constant.

higher potential than observed for CbA5H$^{WT}$ (Fig. 3d). We could fit the above kinetic model to the chronoamperometric data recorded with each of the mutants; the results in Table 1 confirm that the mutations hinder the A$_1$ to H$_{inact}$ conversion and inform about their individual effects on each of the two steps. The L364F and P386L substitutions decrease $k_{inact}$ and increase $k_{react}$ (the P386L substitution also increases $k_{-1}$, Table 1). The A561F substitution only significantly affects the 1$^{st}$ step (A$_1$ to A$_2$): it decreases $k_1$ and increases $k_{-1}$ (Table 1). The double-exchange variant L364F-A561F combines the individual effects of the two single exchanges.

Figure 3e shows that each of the substitutions also decreases O$_2$-resistance. Each of the variants exhibited some residual level of activity just after the first potential step from $-0.8$ V to 0 V, followed by an inactivation that is slower than that of CbA5H$^{WT}$. After buffer exchange and the final step to $-0.8$ V, the residual activities of all single-exchange variants are down to 76–64% and the double-exchange variant merely reaches 34% of the activity recorded prior to O$_2$-exposure, compared to 100% for CbA5H$^{WT}$ (also in Fig. 3b). These results show that resistance to O$_2$ correlates with the overall rate of formation of the inactive state

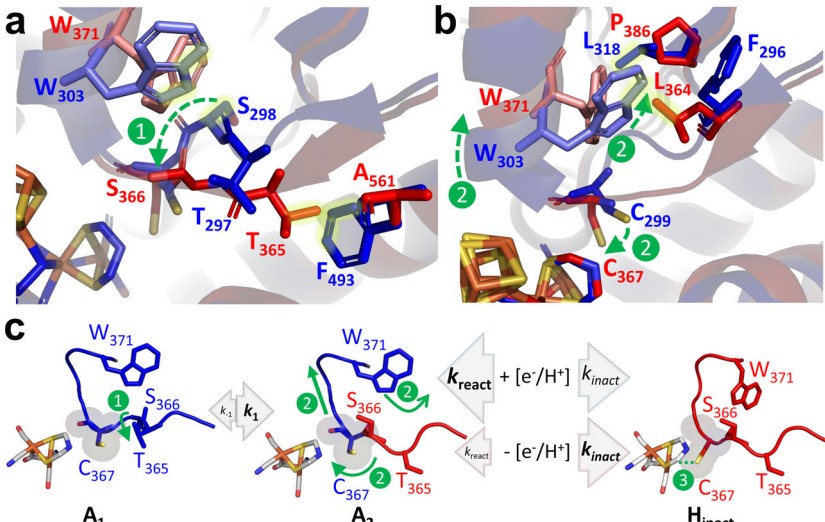

**Fig. 5 Mechanism for the reversible formation of the $O_2$-resistant $H_{inact}$ state in CbA5H$^{WT}$. a, b** Structural alignment of CpI (state $A_1$, blue) and CbA5H$^{air}$ ($H_{inact}$; red) focusing on the TSC-loop. Residues labeled in red determine the structural rearrangement from $A_1$ to $H_{inact}$. Corresponding CpI positions are shown in blue (yellow glow indicates clashes). **c** Illustration of the AAI mechanism. Main steps (1–3) in the transition from $A_1$ (blue) to $H_{inact}$ (red) via $A_2$ (blue-red) are marked in green, corresponding to structural rearrangements (1-3) shown in a-b. **1:** Partial shift of the TSC-loop, including T365, initiating the $A_1$ to $A_2$ transition. **2:** W371-translocation and α-helix uplift, dragging along C367 closer to $Fe_d$ (green arrows) **3:** Binding of C367 to $Fe_d$ (dotted green line) and oxidation of the H-cluster yields the inactive but $O_2$-resistant $H_{inact}$ state. Arrow and font sizes reflect relative rates of $k_1$ and $k_{-1}$ or $k_{inact}$ and $k_{react}$, in CbA5H, which define the dynamic equilibrium between $A_1$, $A_2$, and $H_{inact}$. Under reductive conditions ($+e^-/H^+$) the $k_{inact}/k_{react}$ ratio favors $A_2$, whereas $H_{inact}$ accumulates under oxidative conditions ($-e^-/H^+$).

under oxidizing conditions, which increases in the order WT < L364F ≈ P386L ≈ A561F < L364F–A561F < C367D.

Monitoring the IR-signatures of the H-cluster confirms that the residual $O_2$-resistance of the variants results from the formation of a residual fraction of $H_{inact}$ (Fig. 3f and Supplementary Figure 13). Prior to $O_2$-treatment, all variants show the signature of the $H_{ox}$ state, with minor contributions from $H_{ox}$-CO. However, unlike the case of CbA5H$^{WT}$ (Fig. 3c), these signals disappear upon exposure to air. The only remaining signals after $O_2$-treatment result from $H_{ox}$-CO and a small fraction of $H_{inact}$ which is just above noise level for L364F and P386L, and barely visible for L364F-A561F, thus contrasting with the clear and homogenous $H_{inact}$ spectrum of CbA5H$^{WT}$.

We therefore conclude that each of the three substitutions that were expected to alter the flexibility of the TSC-loop slows down oxidative inactivation, limits the extent of $H_{inact}$-formation and, consequently, decreases the resistance of CbA5H to $O_2$. The exchanges have distinct effects on the two steps of the inactivation process (Table 1), and this information is used hereafter to elucidate the conformational change occurring upon formation of $H_{inact}$.

In summary, we deduce the events leading to oxidative inactivation and accumulation of $H_{inact}$ according to the AAI model (Table 1 and Fig. 5) as follows.

The first step of the conformational change (from state $A_1$, as observed in the structure of CpI, to $A_2$) cannot move C367 significantly closer to $Fe_d$ since $A_2$ is still active. We consider likely that the $A_1/A_2$ transition which involves a turn of the segment including CbA5H-T365 and -S366 in the TSC-loop (equivalent to T297 and S298 in CpI) (Fig. 5a, c, step 1), consistent with this first step being slowed only by the A561F substitution (Table 1) which induces a steric clash with T365 of CbA5H$^{air}$ (Fig. 5a). This movement of T365 and S366 requires a reorganization of the H-bond network that stabilizes TSC loop configuration $A_1$, including a disruption of the H-bond contact between the N-atom of the indole ring and the carbonyl oxygen

of T365 (Supplementary Fig. 15) and provides the necessary space for the subsequent translocation of W371 (see clash between Trp (red) and Ser (blue) in its original position of state $A_1$ in Fig. 5a) and of the alpha helix that bears it (Fig. 5b, c; step 2). The substitution L364F favors state $A_2$ over $H_{inact}$ (Table 1) because it removes the slight steric clash between L364 and the W371 in state $A_2$ (Fig. 5b, blue Trp). The conformational shift of W371 in step 2 should be slowed when P386 is replaced with leucine (see clashes between W371 of CbA5H$^{air}$ and L318 of CpI in Fig. 5b), which is consistent with the transition from $A_2$ to $H_{inact}$ being disfavored in variant P386L (Table 1). Step 2 brings C367 close enough to $Fe_d$ to allow its coordination (Fig. 5c, step 3), which should favor cluster oxidation and deprotonation (probably of the bound cysteine). This oxidation step finally locks down the enzyme in the $H_{inact}$-state.

The protective function of the thiolate group described here is analogous to that of exogenous sulfide demonstrated for several standard [FeFe]-hydrogenases[30]. Sulfide, the main product of the sulphate metabolism of *D. desulfuricans*, could be trapped near the H-cluster and be quickly relocated under oxidative conditions to the open coordination site of $Fe_d$ in order to shield it against $O_2$-attack. To make use of this effect, the presence of extrinsic sulfide as a potential-dependent H-cluster inhibitor would be mandatory. Likewise relying on the capability of a sulfur ligand to reversibly occupy the substrate coordination site of the H-cluster, CbA5H adopts the same protected state as DdH but independent of the presence of external $S^{2-}$. The intrinsic security cap may have been the consequence of evolutionary pressure to cope with incidental $O_2$-exposure in the absence of abundant sulfide.

Our study exemplifies how amino acid residues that are located far from the active site, still may determine the ability to undergo a conformational change which controls chemistry in the 1st coordination sphere, allowing e.g. the enzyme to resist $O_2$ attack. Enzymes benefit from the multiplicity of structural and functional features of the protein scaffold in ways that are far more elaborate

than merely providing a cofactor cavity that stabilizes and tunes the properties of the active site, and substrate/product pathways. The intrinsic cofactor protection mechanism of CbA5H is an impressive showcase for the level of complexity that such additional contributions may reach under corresponding evolutionary pressure.

## Methods

**Protein preparation**. The cDNA used for heterologous overexpression of CbA5H was codon optimized for *Escherichia coli* strain K12 and synthesized with a C-terminal spacer and strep-tagII sequence[27] before being cloned into the pET21b vector, where gene expression is controlled by the T7 promotor. QuikChange PCR was carried out to generate the constructs for the expression of site directed mutagenesis (SDM) variants using mismatch primers (Supplementary Table 3)[18]. Expression constructs were characterized via sequencing.

Protein expression, purification, and in vitro maturation were executed as described before[6,18]. Briefly, apo-forms of CbA5H (wild-type and variants) and CpI lacking the [2Fe2S]-subcluster ([2Fe]$_H$) of the H-cluster were expressed in *Escherichia coli* BL21 (DE3) $\Delta iscR$[42]. Affinity chromatography was applied to isolate the protein under strictly anaerobic conditions in a glove box (Coy Laboratory Products, USA). The purity of the isolate was verified by SDS-polyacrylamide gel electrophoresis and protein concentration was determined via Bradford assay. The purified apo-proteins were incubated with artificially synthesized [2Fe]$_H$-mimic (Fe$_2$[μ-(SCH$_2$)$_2$NH] (CN)$_2$(CO)$_4$[Et$_4$N]$_2$) to reconstitute active protein and excess of the [2Fe]$_H$ mimic was removed by size-exclusion chromatography[43]. The [2Fe]$_H$ mimic used for in vitro maturation was synthesized as described before[44].

**In vitro activity assay**. To perform H$_2$-production activity assays, 400 ng enzyme was mixed with 100 mM NaDT (sodium dithionite) as sacrificial electron donor and 10 mM MV (methyl viologen) as electron mediator in 100 mM K$_2$HPO$_4$/KH$_2$PO$_4$ buffer (pH 6.8). The sealed reaction vessel was kept at 37 °C in a shaking incubator (100 rpm)[6,18,43]. The evolved H$_2$ was quantified by analyzing the headspace of the reaction tube via gas chromatography (Shimadzu). To monitor the pH-dependent H$_2$-production activity, the phosphate buffer was substituted with the respective pH-selective buffers (pH 5 to pH 9)[6,18]. Each measurement was done at least three times (see mean values and standard deviations in Supplementary Figure 7).

**Crystallization and structure determination**. Despite the strictly anaerobic handling of CbA5H$^{WT}$ during purification and maturation, protein crystallization was done aerobically (under air). CbA5H crystals were obtained using the hanging drop vapor diffusion method. 2 μl protein (20–40 mg/ml) was mixed with 2 μl reservoir solution, containing 0.1 M HEPES pH 7.5 (4-(2-hydroxyethyl)-1-piper-azineethanesulfonic acid), 0.4 M MgCl$_2$ and 26–28% PEG400 (polyethylene glycol of molecular weight c.a. 400 Da). The crystals were grown at 18 °C and reached full size within one week. Before flash freezing in liquid N$_2$, the crystals were soaked for a few seconds in cryobuffer, being a premix of reservoir solution and 100% glycerol at a 1:1 ratio. X-ray diffraction data were collected on beamline ID29 at the European Synchrotron Radiation Facility (ESRF, Grenoble, France) at 100 K. Diffraction images were processed with XDS and the processed data were reduced and combined by running XSCALE[45]. Molecular replacement was carried out in Phaser[46] to obtain the phase. Initially, no satisfactory solutions were obtained by using either of the three homologous structures CpI (4XDC), DdH (1HFE) or HydA1 (3LX4). However, a manually trimmed model of CpI (4XDC), in which the non-conserved part was removed based on a sequence alignment, yielded a solution with TFZ (translation function Z) and LLG (log likelihood gain) scores of 14.7 and 137, respectively. Subsequently, Phenix Autobuild was used to generate the model[47]. The first round of Autobuild resulted in an improved model with an R$_{free}$ of 0.38. The model was then further optimized, using alternative rounds of interactive building in Coot[48] and automatic refinement in Phenix. Group occupancy was refined for the iron sulfur clusters. The final model and structure factor were deposited at RCSB protein data bank (PDB ID: 6TTL). Statistics for data collection and refinement are summarized in Supplementary Tables 1, 2. To verify placement of critical parts of the model, simulated annealing omit maps were calculated in Phenix after removing the relevant parts from the input model.

**ATR-FTIR-Spectroscopy**. FTIR-spectra were collected on a Bruker Tensor 27 spectrometer, equipped with a BioATR cell II (Harrick) harboring a double-reflection ZnSe/Si crystal. Prior to all measurements, the sample (15 μL) was dried at 30 °C on the ATR crystal under N$_2$ (10 L/min) to enrich the oxidized ready state H$_{ox}$. A semi-dried protein film was usually obtained after 10 minutes. The film was re-humidified by purging with an aerosol (Tris buffer, pH 8) and the temperature was decreased to 25 °C before the measurements were started (2 cm$^{-1}$ resolution)[49]. The recorded spectra were baseline corrected via OPUS (Bruker GmbH) and then imported and plotted by Origin (Origin Lab). For each enzyme variant, the experiment was repeated two times, showing no significant deviations.

## Electrochemical experiments

*Cyclic voltammetry in Fig. 3 and chronoamperometry in Fig. 4*. Cyclic voltammetry and the amperometry experiments employed to elucidate the kinetics of the conformational change leading to H$_{inact}$-formation were carried out in a JACOMEX glovebox filled with N$_2$ in a thermostated (T = 5°C) electrochemical cell with two compartments. The main compartment contained the rotating working electrode (pyrolytic graphite edge, diameter ~3 mm, typical rotation rate 3 krpm, mounted on a Pine MSR rotator) and the counter electrode (a platinum wire) as well as a tube for constant H$_2$-bubbling. Unless stated otherwise, the main compartment was filled with a mixed buffered solution (consisting of 5 mM of each MES, HEPES, sodium acetate, TAPS, and CHES and 0.1 M NaCl). The main compartment was connected via a Luggin capillary to a second compartment, containing the calomel reference electrode in 0.1 M NaCl. All potentials are quoted with respect to the standard hydrogen electrode, calculated using $E_{SHE} = E_{calomel} + 0.244$ V.

To prepare the protein films, the pyrolytic graphite "edge" (PGE) rotating disc working electrode was polished with an aqueous alumina slurry (Buehler, 1μm), rinsed, then painted with 0.5 μl of an enzyme solution (~ 5 mg/mL in 100 mM Tris-HCl, pH 8 with 2 mM NaDT) and let dry for 2 min. The measured catalytic currents were between 1- and 10 μA, so that mass transport towards the rotating electrode was not limiting[50].

CVs and chronoamperometric experiments were recorded with the program GPES and analyzed with the open source program QSoas[41]. Chronoamperometric experiments such as those presented in Fig. 4a were started and ended at a potential 30 mV below the equilibrium potential (pH 7: −0.456 V vs SHE, pH 8.5: −0.546 V, pH 10: −0.636 V) to ensure complete activation of the enzyme film. The intermediate steps at oxidative potential were of 100 mV amplitude. Before analyzing the H$_2$-oxidation current, we subtracted the background capacitive current recorded in a control experiment without enzyme.

*Chronoamperometry in Fig. 3*. Experiments were carried out in an anaerobic chamber (Coy Laboratory Products, USA) under an atmosphere of 98% N$_2$ and 2% H$_2$. A PalmSens potentiostat (PalmSens 4) was used, controlled by the PalmSens software PSTrace 5.2. The gastight electrochemical cell was water jacketed to control the temperature and a PGE rotating disc electrode was used as a working electrode and controlled by a rotator (Autolab). The reference electrode (Ag/AgCl, 3.5 M KCl) was kept in a non-isothermal side arm, being connected to the main cell compartment by a Luggin capillary. Platinum wire was used as a counter electrode. The reference potential was converted to the standard hydrogen electrode (SHE) scale, using the correction $E_{SHE} = E_{Ag/AgCl} + 0.205$ V at 20 °C. For each experiment, defined gas flow rates (Air liquid, Germany) were adjusted, using mass flow controllers (Sierra Instruments). Enzyme films were prepared by drop-casting 3 μL of a 10 μM hydrogenase sample. After 3 minutes, the electrode was rinsed with MQ water to remove all unbound enzyme molecules. Prior to the potential step experiment, a cyclic voltammogram was recorded for each enzyme film (5 °C, pH 7, 1000 rpm electrode rotation, 100% H$_2$ atmosphere, from −1 to 0 V, using a scan rate of 20 mV/s). For the chronoamperometric experiments, a potential of −0.8 V vs SHE was applied for 200 s (Phase 1, t = 0 s) to monitor H$^+$ reduction activity. A potential step to 0 V vs SHE (phase 2, t = 200 s) was followed by the injection of 50 μM O$_2$ (using air-saturated buffer, phase 3, t = 480 s). A subsequent five-fold buffer exchange (5 mL cell volume, 50 mL volume exchange buffer, incubated at 5 °C) was performed to reestablish anaerobic conditions (phase 4, t = 900 s) prior to the final potential step back to −0.8 V vs SHE (phase 5, t = 1300 s) which was required to measure the residual H$^+$ reduction current. For each enzyme variant the experiment was repeated at least 4 times, showing very similar results.

**Reporting Summary**. Further information on research design is available in the Nature Research Reporting Summary linked to this article.

## Data availability
The coordinates and structure factors of CbA5H$^{air}$ have been deposited in protein data bank (PDB) under PDB-ID: 6TTL All data are available in the main text or the supplementary materials. Further data supporting findings of this study are available from the corresponding authors upon reasonable request. Further publicly available datasets used or indicated in this study comprise PDB-ID: 4XDC, PDB-ID: 3LX4 and PDB-ID: 1HFE. Source data are provided with this paper.

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

## Acknowledgements

We thank the staff at beamline ID29 (ESRF, Grenoble) for technical support during X-ray data collection. M.W., J.D., and T.H. gratefully acknowledge financial support from the Volkswagen-Stiftung (Design of [FeS]-cluster containing Metallo-DNAzymes (Az 93412)), from the Deutsche Forschungsgemeinschaft (DFG, German Research Foundation) under Germany´s Excellence Strategy – EXC 2033 – Project number 390677874 and the DIP Programme (LU 315/17-1). T.H. and E.H. also received funding within the DFG Research Training Group GRK 2341: (Microbial Substrate Conversion). The French authors are supported by the CNRS, Aix Marseille Université, Agence Nationale de la Recherche and the Excellence Initiative of Aix-Marseille University - A*MIDEX, a French "Investissements d'Avenir" programme (ANR-12-BS08-0014, ANR-14-CE05-0010, ANR-11-IDEX-0001-02). The Italian authors are supported by University of Torino funding RICERCA LOCALE and wish to acknowledge the DAAD program for supporting collaboration with the Department of Plant Biochemistry, Ruhr-Universität Bochum and Erasmus traineeship for collaboration with CNRS.

## Author contributions

T.H., M.W. and C.L. conceived and supervised the project, which was initiated during a collaboration with G.G and F.V. about the structure of CbA5H. C.L., C.F. and V.F. were responsible for recording cyclic voltammetries of the protein and kinetic modeling. A.R. crystallized the protein and J.D. and E.H. solved the structure. J.D., M.W. and E.H.

performed structural analysis. M.W. and J.D. designed the mutagenesis variants. L.S. and J. J. purified the protein and measured the activity. M.W., O.L., J.J. and J.D. carried out the chronoamperometry experiment. A.R. and L.S. performed the ATR-FTIR experiments. U.P.A. synthetized the $[2Fe]_H$ complex for in vitro maturation. M.W., C.L., T.H., J.D. and C.F. wrote the manuscript with input from other coauthors. All authors discussed and commented on the manuscript.

## Competing interests

The authors declare no competing interests.
