## [Peer Review File · Nature Communications]

REVIEWER COMMENTS

Reviewer #1 (Remarks to the Author):

The manuscript "A safety cap protects hydrogenase from oxygen attack" by Winkler et al describes the protection mechanism which is not fully understood so far in the case of [FeFe] hydrogenases. [FeFe] hydrogenases have long been known as extremely oxygen sensitive enzyme in the active form. Many attempts had been performed to elucidate the O₂ protection (or destruction) mechanism of [FeFe] hydrogenases, but no one succeeded yet.

Although the IR data and electrochemical data of the series of the variants as well as the kinetic analysis very clearly showed the importance of cysteine367, I doubt the conclusion elucidated by the crystal structures because of the lack of quality of the electron density data. Unless the crystallographic data improves, I would recommend sending to the more specific journals.

Comments:

1. The bond distance between the sulfur atom (Cys367) and the distal Fe atom is slightly longer (3 Å) than that of the standard Fe-S bond. The protonated sulfur atom should be considered. Very recently Corrigan et al reported that the effect of the cysteine residue for O₂ protection in the same enzyme (JACS 2020, doi:10.1021/jacs.0c04964). It should be carefully compared these results.

2. The structural analysis in case of the translational Non-Crystallographic Symmetry (tNCS) is usually very difficult. The methods to solve this problem is useful for readers and the details should be included in the materials and methods section. It seems that the space group P4₁2₁2 is a higher symmetry. Did you calculate in case of the space groups P1 or others?

3. It is better to show the molecules in the asymmetric unit to make clear the disordered domain is not artifact by the crystallographic packing.

Reviewer #2 (Remarks to the Author):

I have very much appreciated the depth and range of analysis the authors have brought to this study of inactivated/protected forms of the hydrogenase H-cluster protein and understand their desire to present the mechanism as novel/a paradigm of remote control of site reactivity in certain hydrogenases. The discussion and conclusions seem well-founded. The 3.1 Å S - Fe(d) distance prompts the question as to whether anionic thiolate or neutral thiol is the ligating moiety. In the context of exploiting the hydrogenases technologically, understanding protection mechanisms against O₂ damage are very important. However, there is much data presented and the key message somewhat loses its impact, it could be substantially shortened before publication which I recommend.

C J Pickett

Reviewer #3 (Remarks to the Author):

The authors present a comprehensive structure-function analysis of the unusual oxygen tolerance behaviour displayed by FeFe hydrogenase CbA5H. Using a combination of electrochemistry, IR spectroscopy, site directed mutagenesis and x-ray structure determination the authors show a loop next to the active site (TSC loop) adopts a novel conformation relative to other hydrogenases allowing C367, part of the proton relay, to bind directly to the active site thus impeding access to diatomic gases. This is a double edged sword, however, since although it protects against oxidative degradation it renders the enzyme inactive. The inactivation is reversible and appears to stem

from several non-conserved residues in the wider active site. Electrochemical, kinetic and spectroscopic analysis allow the authors to propose a plausible mechanism for conformational changes within the active site leading up to formation of the Hinact state.

Overall I found the manuscript to be very detailed analysis with experiments generally performed to a high standard.

That said I do not think it is good practice to omit large regions of the protein from the model where there is interpretable electron density. I appreciate that it did not model as well as the rest of the H- or F-domains, but something really should be included even if it is only a polypeptide backbone. The authors themselves state that inclusion of backbone for this region lowered the R-factors by several percent. While I am aware producing low R-factors is not the point of refinement, but they do indicate a better agreement between F_{obs} and F_{calc} and hence an improvement of phase estimates and consequently maps. While it may not affect the outcome of the discussion and conclusion of this manuscript I would strongly encourage the authors to submit a more complete model as soon as possible.

Could S-SAD be used a method for identifying the Cys/Met residues in the SLBB domain to help locate side chains?

In the manuscript anomalous difference maps are referred quite a lot, which data was used to calculate these? Was a specific data set, for example, at the FeK-edge used to calculate these or are they simply differences present in the 0.976 Å data set used for structure refinement?

Were there any clues, for example, in the normalised temperature factors of the TSC loop that would suggest an enhanced flexibility of this region in CbA5H relative to other FeFe hydrogenases?

How do the authors reconcile the very low Hinact signals in the IR spectra with the differing levels of activity the different variants show (Figure 3E and F). The electrochemistry shows retention of 76-64% of WT activity after oxygen exposure, yet the Hinact signals are barely present. Wouldn't retention of 64% activity suggest 64% of the active site remained intact meaning one would expect the IR signal to be ~64% of that of WT? Could the authors please clarify this apparent discrepancy?

I feel the proposed mechanism is plausible, ideally the structure of the reduced protein would also be presented to confirm the proposed motions, but I assume reduced protein would not crystallise? I am not insisting on seeing the reduced structure, but at least some sort of comment should be included in the manuscript.

In the mechanism the motion of the tryptophan is described as being impeded by steric clashes with L318/386, however the N atom of the indole ring makes a good hydrogen bond with the carbonyl oxygen of T297 of CpI that switches to the carbonyl of S266 in CbA5H. This H-bond to T297 could also impede motion in CpI and its exchange to S266 drive translocation of W371 and its associated helix in CbA5H. Could the authors please comment on this?

As a blue skies question, have the reciprocal variants been made in CpI to attempt to introduce oxygen tolerance into this enzyme?

Minor points

How was cluster occupancy estimated?

Lines 97-98 – The authors mention the electron density shows the H-cluster structure to be unchanged, presumably this is an inference from the spectroscopy rather than the electron density since the resolution is too low to state this with confidence?

Answers to Reviewer comments

Reviewer #1:

The manuscript “A safety cap protects hydrogenase from oxygen attack” by Winkler et al describes the protection mechanism which is not fully understood so far in the case of [FeFe] hydrogenases. [FeFe] hydrogenases have long been known as extremely oxygen sensitive enzyme in the active form. Many attempts had been performed to elucidate the O₂ protection (or destruction) mechanism of [FeFe] hydrogenases, but no one succeeded yet.

R1, comment 1

Although the IR data and electrochemical data of the series of the variants as well as the kinetic analysis very clearly showed the importance of cysteine367, I doubt the conclusion elucidated by the crystal structures because of the lack of quality of the electron density data. Unless the crystallographic data improves, I would recommend sending to the more specific journals.

Answer:

We agree with referee 1 that the resolution of the CbA5H^{air} structure is only moderate, but this does not weaken our results. We took great care in restricting our model to the reliable structure parts which comprise the entire H-domain. As our conclusions for reversible H_{inact} formation solely focus on the H-cluster environment located in the center of the H-domain, we are confident that the arguments made here are valid. The simulated annealing omit maps shown in Fig. 2 and Supplementary Fig. 3, which were calculated to remove possible model bias, also support the new conformation of the loop in the H_{inact} state. To further validate our findings, we checked several additional crystallographic datasets obtained from different batches of purification, crystallization and diffraction experiments. Even though these data were of slightly lower quality, they all support the shifted conformation of the loop region. We therefore believe that the structural conclusions drawn from X-ray crystallography are an important additional observation to support our scientific argument.

Reviewer 1 suggests us to compare our results with the mostly theoretical data presented in the study of Corrigan et al., 2020 which was published after our initial submission (see comment 3 below). Therein, the authors suggest that Cys-binding to Fe_d would require the movement of a flexible loop.:

*“The two best-fitting DFT models of the H_{inact} state, L = OH⁻ and S⁻-Cys, are consistent with this scenario. For the former to take place, a readily available water molecule must exist in the active-site pocket. **For the latter, the Cys367 residue***

near Fe_d must reside on a very mobile flexible loop and have a high probability of rotating its side chain toward the open coordination site to afford coordination.

As our CbA5H^{air} structure clearly supports such a shift of the TSC loop which relocates Cys367 close to Fe_d, our findings are fully in line with the conclusions drawn by Corrigan and coauthors. Our work is original as it combines structural, kinetic and spectroscopic characterizations of a number of variants in order to quantitatively understand the reversible conformational change that protects CbA5H from oxygen.

R1, comment 2

The bond distance between the sulfur atom (Cys367) and the distal Fe atom is slightly longer (3 Å) than that of the standard Fe-S bond. The protonated sulfur atom should be considered.

Answer:

We agree with reviewer 1 and 2 that the distance between S and Fe_d is longer than expected for a typical Fe-S bond although bond length determination may not be very accurate at this moderate resolution. One explanation already mentioned in our manuscript might be that this bond is very weak, which reasonably explains the rapid transitions between H_{ox} and H_{inact}. The loosely attached relocated cysteinate nevertheless blocks the access to the open coordination site at Fe_d. The pH dependence of the rate of reactivation (derived from electrochemically monitored anaerobic oxidative (in)activation) shows that the full oxidation of the H-cluster that leads to H_{inact} is a strictly coupled one-electron one-proton process. We consider it likely that this proton comes from the cysteine and that in H_{inact}, C367 binds Fe_d in a deprotonated cysteinate state. A recent publication about the binding of H₂S to the same Fe_d site suggests that deprotonation of the H₂S ligand occurs when the cluster is fully oxidized (*Rodriguez-Macia et al., ACIE June 2020 10.1002/anie.202005208*). To clarify this in our manuscript, we have revised the following sentences:

- pages 13-14, lines 263-268:

“The fact that k_{inact} is potential-independent points to a classical “CE” mechanism for step 2, in which one or a series of chemical step(s) (“C”) precede(s) proton-coupled oxidation (“E”). Strong coupling implies that the sites of deprotonation and oxidation are very close to one another. We therefore consider it likely that this deprotonation occurs from the nitrogen atom of the azadithiolate bridge, or from the bound cysteine, to produce a thiolate ligand.”

- page 18, line 352:

“Step 2 brings C367 close enough to Fe_d to allow the coordination of the thiolate (Figure 5C, step 3), which should favor cluster oxidation **and deprotonation (probably of the bound cysteine)**. This oxidation step finally locks the enzyme in the H_{inact}-state”.

R1, comment 3

Very recently Corrigan et al reported that the effect of the cysteine residue for O₂ protection in the same enzyme (JACS 2020, doi:10.1021/jacs.0c04964). It should be carefully compare these results.

Answer:

The manuscript of Corrigan and coworkers has been published on June 20th (10.1021/jacs.0c04964), well after our initial submission. In that paper, based on only DFT calculations, the binding of a cysteine to Fe_d is only one of two distinct hypotheses. We reach a definite conclusion in our paper, supported by a thorough experimental investigation.

We have now included references to this work in our manuscript wherever appropriate. It is now cited on page 3, lines 67 and 69, and the following sentences have been added:

- page 8, line 163

“**as recently observed in another electrochemical investigation of CbA5H**³¹”,

- and on page 5, line 111-113:

“**This observation is consistent with a recently published hypothesis according to which the loop that bears C367 in CbA5H may be flexible enough to allow the binding of the cysteine to Fe_d**³¹”

R1, comment 4

The structural analysis in case of the translational Non-Crystallographic Symmetry (tNCS) is usually very difficult. The methods to solve this problem is useful for readers and the details should be included in the materials and methods section. It seems that the space group P41212 is a higher symmetry. Did you calculate in case of the space groups P1 or others?

Answer:

Indeed, presence of tNCS impedes structure solution. In our case, the automated tNCS treatment in PHASER (used for MR) was sufficient to allow placement of the two related copies in the ASU (which represent a “simple” case of tNCS). This information was already available in the material and methods section.

Concerning the space group, of course we doublechecked and analyzed the data both, in P1 (as the most general case) and in all relevant tetragonal spacegroups.

We now added this information to the text below Supplementary Tab. 1:

As a control, data were also processed in space group P1 and all relevant tetragonal space groups. In these cases, either molecular replacement failed, or crystallographic statistics and map qualities were clearly inferior to space group $P4_2 2_1 2$.

R1, comment 5:

It is better to show the molecules in the asymmetric unit to make clear the disordered domain is not artifact by the crystallographic packing.

Answer:

We showed the asymmetric unit in Supplementary Fig. 2 and also checked the interfaces of the disordered F- and SLBB domains between neighbouring asymmetric units. There are no indications that packing artefacts are the origin for the disordered structure parts. We assume that the connections of the disordered domains with the H-domain are inherently too flexible to sufficiently stabilize the F-and SLBB domains under our crystallization conditions.

Reviewer #2:

I have very much appreciated the depth and range of analysis the authors have brought to this study of inactivated/protected forms of the hydrogenase H-cluster protein and understand their desire to present the mechanism as novel/a paradigm of remote control of site reactivity in certain hydrogenases. The discussion and conclusions seem well-founded.

R2, comment 1:

The 3.1Å S - Fe(d) distance prompts the question as to whether anionic thiolate or neutral thiol is the ligating moiety.

Answer:

We kindly refer reviewer 2 to our answer to R1, comment 2.

R2, comment 2:

However, there is much data presented and the key message somewhat loses its impact, it could be substantially shortened before publication which I recommend. C

J Pickett

Answer:

The key message of our study is to derive the mechanism of reversible H_{inact} formation and to outline the central mechanistic contribution of C367 as far as our experimental approaches allow us to do so. Shortening the text significantly, would mean to skip an entire part. Each experimental section (X-ray structure, protein electrochemistry, ATR spectroscopy and kinetics data) contributes here in a complementary rather than repetitive fashion. We therefore do not share the impression that we present too much data which may distract from the core message or weaken the impact of our study. This is furthermore supported by referees 1 and 3 who ensured their overall positive impression about the detailed analysis and extent of our study.

R1: ‘...the IR data and electrochemical data of the series of the variants as well as the kinetic analysis very clearly showed the importance of cysteine367...’

R3: ‘...Overall I found the manuscript to be very detailed analysis with experiments generally performed to a high standard...’

Reviewer #3:

The authors present a comprehensive structure-function analysis of the unusual oxygen tolerance behaviour displayed by FeFe hydrogenase CbA5H. Using a combination of electrochemistry, IR spectroscopy, site directed mutagenesis and x-ray structure determination the authors show a loop next to the active site (TSC loop) adopts a novel conformation relative to other hydrogenases allowing C367, part of the proton relay, to bind directly to the active site thus impeding access to diatomic gases. This is a double edged sword, however, since although it protect against oxidative degradation it renders the enzyme inactive. The inactivation is reversible and appears to stem from several non-conserved residues in the wider active site. Electrochemical, kinetic and spectroscopic analysis allow the authors to propose a plausible mechanism for conformational changes within the active site leading up to formation of the H_{inact} state.

Overall, I found the manuscript to be very detailed analysis with experiments generally performed to a high standard.

R3, comment 1:

That said I do not think it is good practice to omit large regions of the protein from the model where there is interpretable electron density. I appreciate that it did not model as well as the rest of the H- or F-domains, but something really should be included

even if it is only a polypeptide backbone. The authors themselves state that inclusion of backbone for this region lowered the R-factors by several percent. While I am aware producing low R-factors is not the point of refinement, but they do indicate a better agreement between F_{obs} and F_{calc} and hence an improvement of phase estimates and consequently maps. While it may not affect the outcome of the discussion and conclusion of this manuscript I would strongly encourage the authors to submit a more complete model as soon as possible.

Answer:

We understand the concerns of Reviewer 3. However, we tried numerous approaches to make the model as complete as possible and push the limits of resolution. To decrease the disorder, we even collected X-ray data at room-temperature (RT). Several full datasets collected at RT indeed improved the disorder significantly (Wilson B factors 55 vs. 87.5). To achieve a full dataset at the highest possible resolution of 2.9 Å (same as the one presented in the manuscript) with room-temperature data, we merged the data from 5-10 crystals. Nevertheless, it was not possible to model the N-terminal part of the SLBB domain in the resulting maps. The best resolution gained from a single crystal measured at room temperature was 3.2 Å due to severe radiation damage.

Given the suggestion of the reviewer, we again carefully compared the refinement results with and without modelling the backbone of the SLBB domain. Apart from a decrease of R_{free}/R_{work} by 2-3%, the phase errors, FOMs (figure of merit) and overall density did not improve significantly upon modelling the backbone of the SLBB domain. It was still impossible to locate the electron density of individual side chains, underlining the inherent flexibility of the domain. Therefore, we would still prefer not to include the backbone fragments in the final model deposited in PDB. We prefer to clearly illustrate the 2 large regions which were not modelled in the manuscript (Supplementary Fig. 2), rather than presenting the fragments in the form of residue backbone for which we would have to assign all the residues as Ala or Gly, which in the end would rather confuse the readers and users of pdb.

R3, comment 2:

Could S-SAD be used a method for identifying the Cys/Met residues in the SLBB domain to help locate side chains?

Answer:

We thank referee 3 for suggesting this interesting approach to further resolve the N-terminal part of the SLBB domain. Sadly, in our case there is a wide range of

difficulties when applying S-SAD. Usually S-SAD requires crystals of relatively good quality (in most cases better than 2.5 Å) due to the very weak anomalous signals from S. The presence of a large number of iron atoms will complicate quantitative analysis of anomalous scattering of S. Besides, the SLBB domain is inherently less ordered than the H-domain. Finally, only 2 Cys are present in the unresolved part of the SLBB domain. All these facts will make the S-SAD very challenging. Still, we tried to collect a number of datasets at wavelengths of 2 Å to increase the anomalous signals from S. We followed the well-established data collection strategy regarding native SAD phasing mainly based on S (Weinert et al. 2015, doi.org/10.1038/nmeth.3211). Diffraction data were collected at different orientations in high redundancy (4*360°) and fine ϕ -slicing to adequately increase the accuracy of the weak anomalous signal from S. Unfortunately, this approach was not successful. We calculated the anomalous maps from the S-SAD datasets. Even the anomalous densities of Cys and Met in the well-resolved H-domain are not good enough, suggesting that the data quality is not sufficient to apply S-SAD.

As already stated in the methods section, Fe-SAD was attempted as well for experimental phasing, being significantly more powerful in phasing of iron-sulfur proteins than S-SAD. The success of Fe-SAD clearly supports the structural model presented here (see details in the method part).

R3, comment 3:

In the manuscript anomalous difference maps are referred quite a lot, which data was used to calculate these? Was a specific data set, for example, at the FeK-edge used to calculate these or are they simply differences present in the 0.976 Å data set used for structure refinement?

Answer:

We apologize for the misunderstanding, the structural results we presented in the manuscript were based on one dataset only which was collected at 0.976 Å. We processed the same data set in parallel with the Friedel pairs treated differently to see the difference. The statistics including anomalous scattering are now presented in Supplementary Table 1. Even at this wavelength, Fe-atoms still show an anomalous scattering component f'' of about 1.6e, allowing the localization of the FeS-clusters in these maps.

R3, comment 4:

Were there any clues, for example, in the normalised temperature factors of the TSC loop that would suggest an enhanced flexibility of this region in CbA5H relative to other FeFe hydrogenases?

Answer:

If it would concern a structure of the more reduced active state(s) of CbA5H we would agree that the constant shift between A1 and A2 reflected in the translocation of S366 and T365 should be indicated by larger b-factors in parts of the TSC loop compared to the remaining H-cluster environment. In our crystals, we are analyzing the X-ray structure of the H_{inact} state for which the TSC loop is trapped in the fully shifted configuration and therefore is unlikely to exhibit increased b-factor values. A coloration of the H-cluster environment in CbA5H^{air} according to the b-factor supports this conclusion (see Fig. below). To clarify this point, we included this Figure into the SI part as Supplementary Fig. 16 and added the following sentence to the main text at page 10, lines 204-207:

That CbA5H is quantitatively locked in the H_{inact} configuration is not only obvious from the homogenous IR-spectra of O_2 treated enzyme, but also from the low b-factor values observed for the H-cluster environment in CbA5H^{air} (6TTL), including the fully shifted TSC-loop (Supplementary Fig. 16).

Supplementary Fig. 16| Comparison of local b-factor differences in the x-ray structures of Cpl (pdb code: 4xdc) and CbA5H^{air} (6TTL). A+B: Cartoon structure

models of Cpl and CbA5H^{air} colored according to the b-factor of individual structure parts. **C+D:** Close up of the H-cluster and the TSC loop region with residues participating in the O₂-resistance mechanism depicted as stick structures. No significant b-factor deviation is visible in the environment of the H-cluster or the TSC-loop region. The dominating blue color range indicates for both structures likewise low b-factors in the center of the H-domain.

R3, comment 5

How do the authors reconcile the very low H_{inact} signals in the IR spectra with the differing levels of activity the different variants show (Figure 3E and F). The electrochemistry shows retention of 76-64% of WT activity after oxygen exposure, yet the H_{inact} signals are barely present. Wouldn't retention of 64% activity suggest 64% of the active site remained intact meaning one would expect the IR signal to be ~64% of that of WT? Could the authors please clarify this apparent discrepancy?

Answer:

The system conditions applied in our chronoamperometry experiment and during ATR-FTIR spectroscopy differ inevitably foremost in respect of enzyme concentration and O₂ treatment but also concerning other system parameters.

While for chronoamperometry only 3 µl of a 10 µM protein sample was adsorbed on the working electrode, we merely added 50µl of air saturated buffer once to the 5ml buffer volume in an electrochemical cell that is constantly purged with H₂ gas which continually drives out part of the dissolved O₂ from the system. For ATR-FTIR-spectroscopy, we purged 15 µl of a 200µM sample in a semi-dried protein film for 10-15 min with air (30 ml air/min) which is a significantly harsher treatment with undiminished O₂ exposure over a significantly longer period of time. We therefore did not observe the exact same effects in the two experiments, but the qualitative trends and relative effects are similar when the WT enzyme and the mutants are compared.

R3, comment 6

I feel the proposed mechanism is plausible, ideally the structure of the reduced protein would also be presented to confirm the proposed motions, but I assume reduced protein would not crystallise? I am not insisting on seeing the reduced structure, but at least some sort of comment should be included in the manuscript.

Answer:

We certainly agree with reviewer 3. Despite our best efforts, we were not able to

crystallize the protein under anaerobic conditions yet. To emphasize this, we added the following sentence to the manuscript at page 5, lines 100-101: ‘*Since we could not crystallize CbA5H under anaerobic conditions, we compared the structures of CbA5H^{air} and Cpl, a “standard” [FeFe]-hydrogenase.*’

R3, comment 7

In the mechanism the motion of the tryptophan is described as being impeded by steric clashes with L318/386, however the N atom of the indole ring makes a good hydrogen bond with the carbonyl oxygen of T297 of Cpl that switches to the carbonyl of S266 in CbA5H. This H-bond to T297 could also impede motion in Cpl and its exchange to S266 drive translocation of W371 and its associated helix in CbA5H. Could the authors please comment on this?

Answer:

We thank reviewer 3 for this insightful comment. According to the model proposed in our manuscript, the H-bond networks at the TSC loop in Cpl and A1 of CbA5H should be very similar. In both cases the carbonyl-oxygen of the corresponding threonine position (T279^{Cpl}/T365^{CbA5H}) would be engaged in an H-bond contact with Trp (W303^{Cpl}/W371^{CbA5H}). In case of CbA5H, overall lower steric constraints (e.g. due to the significantly smaller residue at position A561^{CbA5H} instead of F493^{Cpl}) enable the reversible transition between states A1 and A2, including the half-turn of the TSC loop comprising T365 and S366. Cpl however, is incapable of switching to state A2 due to the larger steric barriers that fix the TSC loop in its most effective position (please keep in mind that A1 is more active compared to A2 and thus evolutionarily favored in terms of catalytic efficiency). The shift to A2 certainly changes the H-bond pattern in CbA5H which includes the disruption of the H-bond between T367 and W371. The higher level of rotational freedom (due to the translocation of barrier position S366), the steric influence of the likewise relocated position L364 and the new location of the hydroxo-group of S366 may favor the shift of W371 to its second conformation visible in CbA5H^{air}.

We now have added a supplementary Fig. (Supplementary Fig. 15) depicting a comparison between the H-bond networks stabilizing state A1 (corresponding to the configuration in Cpl) and the fully translocated H_{inact} state as observed in the structure of CbA5H^{air}.

To furthermore emphasize the relevance of the H-bond network for stabilizing the different states in the transition process, we revised the sentence on page 17, line 341-346 as follows:

'This movement of T365 and S366 requires a reorganization of the H-bond network that stabilizes the A1 configuration of the TSC loop, including a disruption of the H-bond contact between the N atom of the indole ring and the carbonyl oxygen of T365 (Supplementary Fig. 15) and provides the necessary space for the subsequent translocation of W371 (see clash between Trp (red) and Ser (blue) in its original position of state A1 in Figure 5A) and of the alpha helix that bears it (Figure 5B-C; step 2).'

Fig. 15| H-bond contacts stabilizing the states A₁ and H_{inact} in CbA5H. **A:** Putative H-bond network stabilizing the configuration of the TSC loop in Cpl which likely corresponds to state A₁ in CbA5H. Water molecules contributing to the H-bond network are depicted as green balls. **B:** H-bond network stabilizing the rotated TSC loop configuration, initiating the shift of W371 and of the corresponding alpha helix, which leads to the translocation of C367. Unfortunately, the moderate resolution of CbA5H^{air} does not permit a localization of the contributing water molecules in the H_{inact} state of CbA5H^{air}. As the water molecules depicted in panel A are strictly conserved, we can only speculate about their positions in CbA5H according to available space and potential H-bond partners (broken circles and parentheses).

R3, comment 8

As a blue skies question, have the reciprocal variants been made in Cpl to attempt to introduce oxygen tolerance into this enzyme?

Answer:

This is indeed the basis of an exciting follow-up project, and we are currently planning to work on this, but we cannot comment at this point.

R3, comment 9

How was cluster occupancy estimated?

Lines 97-98 – The authors mention the electron density shows the H-cluster structure to be unchanged, presumably this is an inference from the spectroscopy rather than the electron density since the resolution is too low to state this with confidence?

Answer:

We think that there may be a misunderstanding here. When we say that the overall H-cluster domain is very similar with that of Cpl, we refer to the whole domain, not the H-cluster. We completely agree that the resolution does not allow us to comment on the details of the H-cluster geometry. However, as indicated in the methods part, group occupancy refinement (many cycles) allows us to estimate the occupancies of the iron-sulfur clusters in a relatively accurate way even at this moderate resolution.

REVIEWER COMMENTS

Reviewer #1 (Remarks to the Author):

The revised version of the manuscript (NCOMMS-20-24274A) by Winkler et al answered appropriately my concerns except one issue (quality of the crystallographic data). The methodological procedures were properly carried out. However, quality of the crystallographic data and the electron density map (Fig 2D) are not improved. For example, the electron density map of the side chain of Cys 367 was assigned but next Cys368 was not assigned. Fig S16C shows that the local b-factor (the region I363-W371) were converged but the electron density map of the peptide chain between L364 and T365 can not follow in the omit map (Fig2D). These indicate that the 2.9 Å resolution data analysis is not enough to assign the Cys 367 sulphur atom. The authors, therefore, should perform the higher resolution data analysis to obtain the reliable results.

Reviewer #2 (Remarks to the Author):

The authors have addressed my comments and I am happy to recommend publication.

Reviewer #3 (Remarks to the Author):

I would like to thank the authors for their clear responses to my questions and the subsequent amendments to the manuscript.

I think the information in the responses to question 1 and 2 regarding modelling the SLBB domain and S-SAD maps would be useful to add as additional methods to the supplementary data file, since the authors were clearly very diligent in structure determination and there is clearly a lot of valuable "know-how" in these paragraphs.

Once this is added I would be happy to recommend publication

Reviewer #1

Reviewer 1, comment 1:

The revised version of the manuscript (NCOMMS-20-24274A) by Winkler et al answered appropriately my concerns except one issue (quality of the crystallographic data). The methodological procedures were properly carried out. However, quality of the crystallographic data and the electron density map (Fig 2D) are not improved. For example, the electron density map of the side chain of Cys 367 was assigned but next Cys368 was not assigned.

Answer:

As indicated before, the structure is definitely well enough resolved to support all the discussion points in the manuscript. As for Cys368 (corresponding to C300 in CpI, see sequence alignment in Supplementary Fig. 4), this amino acid is highly conserved among [FeFe]-hydrogenases as it coordinates the $[4Fe]_H$ cluster. Accordingly, a significant change for C368 was neither expected nor observed in the structure of CbA5H^{air} and consequently, C368 was not discussed in the manuscript. Its structural assignment has now been added as part of an additional Supplementary Figure (Supplementary Fig 17) in the second revision.

Supplementary Fig. 17| Structural comparison between CbA5H and Cpl. Panels **a** and **c** show simulated omit maps (Fo-Fc) for CbA5H^{air} and Cpl contoured at 2 and 3 σ , respectively over the corresponding stick structures. Panel **b** depicts a superposition of stick models for CbA5H^{air} and Cpl, including the H-cluster, the loop region and its local environment. Carbon atoms were colored in yellow and green for CbA5H^{air} and Cpl respectively.

Reviewer 1, comment 2:

Fig S16C shows that the local b-factor (the region (I363-W371) were converged but the electron density map of the peptide chain between L364 and T365 cannot follow in the omit map (Fig2D).

Answer:

For the sake of clarity, we depicted only a selected number of residues in panels D and E of Fig.2. To clarify this, we changed the legend as given below:

Original version:

...(E) Structural alignment, depicting conformational differences between CbA5H^{air} (yellow) and Cpl (white) near the [2Fe]_H-cluster and side chains of amino acids which influence anaerobic inactivation, O₂-resistance and H_{inact} formation. (F)...

Revised version:

...(E) Structural alignment, depicting conformational differences between CbA5H^{air} (yellow) and Cpl (white). Panels **D** and **E** focus on the [2Fe]_H-cluster and side chains of amino acids which influence anaerobic inactivation, O₂-resistance and H_{inact} formation. (F)...

Compared to other regions in Fig. 2D the omit electron density in the peptide region between L364 and T365 is slightly weaker, but still sufficient to assign the residues and the covalent bond between them. We kindly refer reviewer 1 to Supplementary Fig 17 which clearly shows sufficient electron density between L364 and T365. Regarding the larger and more elaborate electron density map shown in the new Supplementary Fig. 17, we have added a sentence in the caption of Fig. 2D: (see Supplementary Fig. 17 for a more detailed structural comparison between the loop regions of Cpl and CbA5H^{air}).

Reviewer 1, comment 3:

These indicate that the 2.9 Å resolution data analysis is not enough to assign the Cys 367 sulphur atom. The authors, therefore, should perform the higher resolution data analysis to obtain the reliable results.

Answer:

We respectfully disagree with reviewer 1 on this. Supplementary Fig. 3, depicting the stick model derived from a well-defined electron density map, unequivocally illustrates the new loop conformation, including the consequently shifted position of C367 in CbA5H^{air} (now further illustrated in Supplementary Fig. 17), showing that the resolution of our structural data is sufficient to come to this conclusion. As outlined in our previous revision, the results we present here are highly reliable

and reproducible (supported by several crystallographic datasets obtained from different batches of protein purification, crystallization and diffraction experiments). Most importantly, the structural observations are perfectly in line with the results of our thorough mutagenesis study and the corresponding electrochemistry and IR spectroscopy experiments.

Reviewer #2 (Remarks to the Author):

The authors have addressed my comments and I am happy to recommend publication.

Answer:

We thank the reviewer for his/her recommendation.

Reviewer #3 (Remarks to the Author):

I would like to thank the authors for their clear responses to my questions and the subsequent amendments to the manuscript.

I think the information in the responses to question 1 and 2 regarding modelling the SLBB domain and S-SAD maps would be useful to add as additional methods to the supplementary data file, since the authors were clearly very diligent in structure determination and there is clearly a lot of valuable "know-how" in these paragraphs.

Once this is added I would be happy to recommend publication.

Answer:

We thank reviewer 3 for the kind recommendation and agree that this information may be helpful for the readers. We therefore added the following sections to Supplementary Note 1:

However, the phase errors, FOMs (figure of merit) and overall density did not improve significantly upon modelling the backbone of the SLBB domain. The electron density of the sidechains does not allow the assignment of the sequence, underlining the inherent flexibility of the domain.

To further explore the possibilities of modelling the SLBB domain, two different strategies were followed. To avoid potential disorder due to cryocooling, we collected X-ray data at room temperature (RT). Several full datasets collected at RT indeed improved the disorder significantly (Wilson B factors 55 vs. 87.5, Supplementary Table 1). To achieve a full dataset at the highest possible resolution of 2.9 Å (same as the one presented in the manuscript) with RT data, we merged the data from 5-10 crystals. Nevertheless, it was not possible to model the N-terminal part of the SLBB domain in the resulting maps despite improved density in the H-domain. The best resolution gained from a single crystal measured at room temperature was 3.2 Å due to severe radiation damage.

To resolve the missing part of the SLBB domain that contains two cysteine residues, we collected further data sets with high redundancy at longer wavelength (2.0 Å) to make use of the anomalous signal of sulfur atoms for localization and for phase improvement. However, these data did not contain enough information for further modelling, as even the anomalous densities of sulfur atoms in the well-resolved H-domain were not very defined.

REVIEWERS' COMMENTS

Reviewer #1 (Remarks to the Author):

Fig S17 clearly shows the model fitting that can support the conclusion. The revised manuscript satisfy my concerns as the authors said the data is reproducible so that I also support the publication.